DISCOVERY REPORT

# Structures of the Hepaci-, Pegi-, and Pestiviruses envelope proteins suggest a novel membrane fusion mechanism

**Michael R. Oliver◎, Kamilla Toon◎, Charlotte B. Lewis, Stephen Devlin, Robert J. Gifford, Joe Grove◎ \***

MRC-University of Glasgow Centre for Virus Research, Glasgow, United Kingdom

◎ These authors contributed equally to this work.
\* joe.grove@glasgow.ac.uk

The Editors encourage authors to publish research updates to this article type. Please follow the link in the citation below to view any related articles.

## Abstract

Enveloped viruses encode specialised glycoproteins that mediate fusion of viral and host membranes. Discovery and understanding of the molecular mechanisms of fusion have been achieved through structural analyses of glycoproteins from many different viruses, and yet the fusion mechanisms of some viral genera remain unknown. We have employed systematic genome annotation and AlphaFold modelling to predict the structures of the E1E2 glycoproteins from 60 viral species in the *Hepacivirus*, *Pegivirus*, and *Pestivirus* genera. While the predicted structure of E2 varied widely, E1 exhibited a very consistent fold across genera, despite little or no similarity at the sequence level. Critically, the structure of E1 is unlike any other known viral glycoprotein. This suggests that the Hepaci-, Pegi-, and Pestiviruses may possess a common and novel membrane fusion mechanism. Comparison of E1E2 models from various species reveals recurrent features that are likely to be mechanistically important and sheds light on the evolution of membrane fusion in these viral genera. These findings provide new fundamental understanding of viral membrane fusion and are relevant to structure-guided vaccinology.

## Background

Enveloped viruses encode specialised glycoproteins that undergo dramatic conformational change to mediate fusion of viral and host membranes. To date, 3 distinct classes of fusion protein are known: Class-I, as found in, for example, influenzaviruses, retroviruses, and coronaviruses [1–7]; class-II, in flaviviruses, alphaviruses, and bunyaviruses [8–14]; and class-III, in rhabdoviruses, herpesviruses, and baculoviruses [15–18]. Mechanistic conservation suggests each class of fusion protein arose from individual ancient progenitors that underwent genetic transfer within the virome. However, subsequent evolutionary divergence has largely eliminated sequence similarity between the glycoproteins within a given class. Therefore, mechanistic/evolutionary grouping of fusion proteins cannot be inferred from primary sequence and is only revealed by structural analysis. Moreover, determination of the structure of viral glycoproteins in both their pre- and postfusion states is critical to understanding the conformational

**Data Availability Statement:** All AlphaFold structures and associated protein sequences are available here: http://doi.org/10.5281/zenodo.7221315. Underlying numerical data is available in S1 File.

**Funding:** This work was supported by the Wellcome Trust and Royal Society through a Sir Henry Dale Fellowship to JG (Grant Number:107653/Z/15/Z); MRC-University of Glasgow Centre for Virus Research core support from UKRI/Medical Research Council to JG (Grant Number: MC_UU_12014); and a Lord Kelvin Adam Smith Fellowship to JG. SD is the recipient of an MRC PhD studentship (Grant Number: MC_ST_CVR_2019). The funders had no role in study design, data collection and analysis, decision to publish, or preparation of the manuscript. JG and MO received salary support through the Sir Henry Dale Fellowship (Wellcome Trust and Royal Society); CL received salary support UKRI/Medical Research Council; KT received salary support through the Lord Kelvin Adam Smith Fellowship.

**Competing interests:** The authors have declared that no competing interests exist.

rearrangements that drive membrane fusion. This knowledge has informed vaccinology [19–21] and provided insights on fusion in eukaryotes [22].

However, there are viruses for which the mechanism of fusion remains unknown; for example, the Hepaciviruses (e.g., hepatitis C virus (HCV)), Pegiviruses, and Pestiviruses (e.g., bovine viral diarrhoea virus (BVDV)), which are all members of the Flaviviridae family. While orthoflaviviruses (e.g. dengue virus) possess class-II fusion proteins, the current structural understanding of HCV and BVDV suggest distinct and, hitherto, unknown fusion mechanism(s) [23–26]. Hepaci-, Pegi-, and Pestiviruses achieve fusion via the E1 and E2 glycoproteins, which form heterodimers and act in concert. HCV and BVDV E2 are structurally distinct, raising the possibility that they have different fusion mechanisms. Here, we propose a common fusion mechanism across the Hepaci-, Pegi-, and Pestiviruses. This is evidenced by high structural conservation in the glycoprotein E1, revealed by ab initio protein structure prediction. Our comparison of E1E2 models from varied species provide insights on potential mechanism and evolutionary origin.

## Structural conservation of Hepaci-, Pegi-, and Pestivirus E1

We performed systematic genome alignment and annotation to generate matched E1E2 targets for AlphaFold protein structure prediction using the ColabFold platform [27,28] (total of 61 E1E2 targets), yielding high confidence E1E2 dimer models from 11, 3, and 10 Hepaci-, Pegi-, and Pestiviruses, respectively (for a detailed description of AlphaFold modelling, see S1 Text and accompanying S1–S14 Figs). E2 models were structurally distinct between (and, to some extent, within) viral genera, whereas E1 models exhibited high structural similarity (Fig 1A). These observations were supported by unbiased all-against-all analysis of structural similarity using the DALI server [29] (Figs 1B and S15 and S1 Text). E1 has 3 conserved structural units found across the viral genera (Fig 1C): (i) a central antiparallel beta-sheet; (ii) a helical hairpin, which corresponds to the putative fusion peptide (pFP) in HCV [30–34]; and (iii) a transmembrane proximal region.

AlphaFold/ColabFold automatically generates multiple sequence alignments based on homology to the given target sequence [27,28]; protein structure is inferred using the evolutionary relationships revealed in these alignments. Therefore, structural similarity between targets can be driven simply by primary sequence similarity (i.e., 2 closely related target proteins are likely to yield similar structures). Consequently, agreement in structure prediction between multiple homologous targets is not a guarantee of accuracy; they could all be similarly incorrect. However, if targets with little/no sequence similarity yield similar structures, there is more confidence in their veracity. We, therefore, compared the sequence data underlying our various E1E2 structure predictions. Hepacivirus E1E2 models draw heavily on HCV sequences; however, with greater genetic distance (i.e., Pegi- and Pestiviruses), the structural models become increasingly independent. Pestivirus E1E2 models share no overlap in their underlying sequence data with Hepaci- or Pegiviruses (Fig 1D and 1E). Therefore, Hepaci-/Pegi- and Pestivirus E1E2 models are completely independent, providing high confidence that their apparent structural similarity is accurate. The high degree of E1 structural similarity across these genetically divergent viral genera is highly suggestive of a shared mechanistically conserved fusion mechanism.

## Molecular architecture of E1

The predicted structure of E1 exhibits a conserved topology and compact architecture in close apposition to the viral membrane, the approximate location of which is inferred by the position of the E1 transmembrane domain (Fig 2A and 2B). Towards the N-terminus, a central

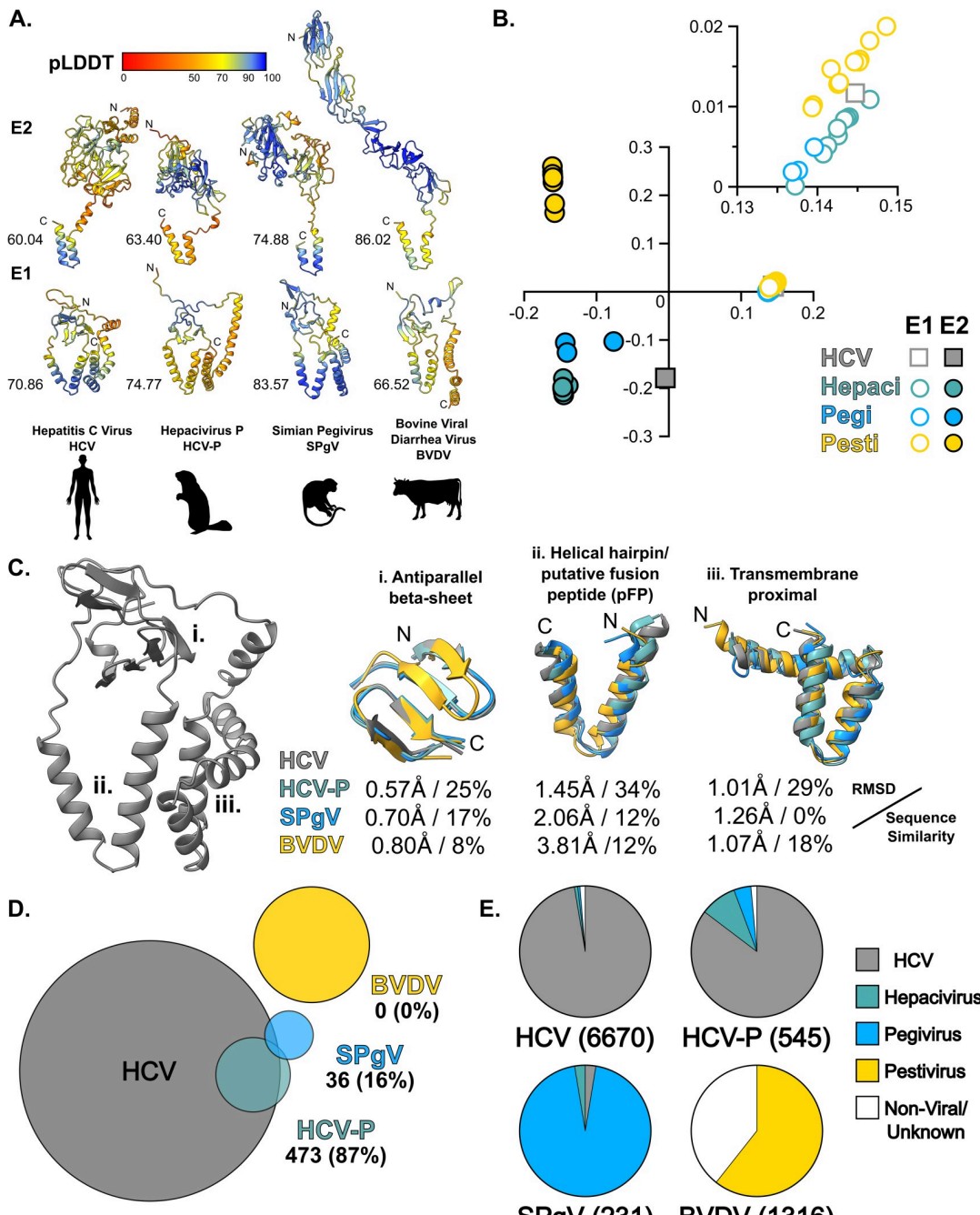

**Fig 1. Structural conservation of E1 across the Hepaci-, Pegi-, and Pestiviruses.** (A) AlphaFold predicted E1 and E2 structures, shown as ribbon diagrams, for HCV, HCV-P, SPgV, and BVDV. Models are colour coded by pLDDT prediction confidence. Host species silhouettes from PhyloPic. (B) Comparison and clustering of E1 and E2 (from 24 species) based on structural similarity through correspondence analysis; the plot is a projection of the first 2 eigenvectors. Inset axes are enlarged to highlight the E1 structural cluster. (C) Consistent features of E1 shared across genera. Left, structure of HCV E1 for context. Ribbon diagrams of overlaid features for HCV, HCV-P, SPgV, and BVDV are accompanied by RMSD and sequence similarity values by comparison to HCV. (D) Venn diagram demonstrating overlap in underlying MSA data used to generate predicted structures for HCV, HCV-P, SPgV, and BVDV. The number of sequences (and percentage) overlapping with the HCV MSA is provided. (E) The constituent sequences from each MSA, classified by source. The number of sequences in each MSA is provided in parenthesis. Underlying numerical data are available in S1 File. BVDV, bovine viral diarrhoea virus; HCV, hepatitis C virus; HCV-P, Hepacivirus-P; MSA, multiple sequence alignment; RMSD, root mean square deviation; SPgV, Simian Pegivirus.

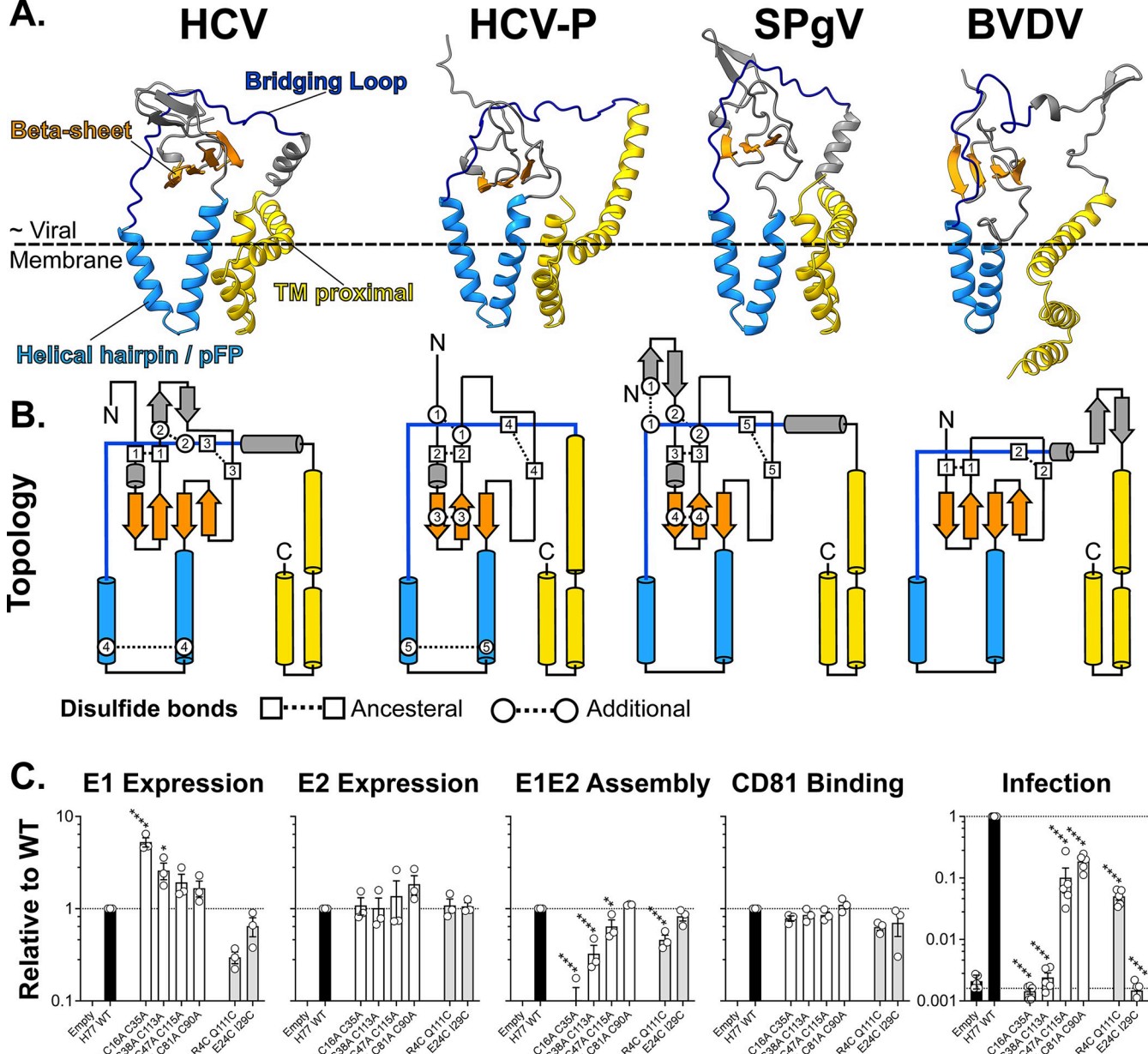

**Fig 2. The molecular architecture of E1. (A)** Ribbon diagrams of predicted E1 structures for HCV, HCV-P, SPgV, and BVDV; features are colour coded as shown on the HCV structure, left. The approximate location of the outer leaflet of the viral membrane is inferred by the positions of the E1 transmembrane domain. **(B)** Topology diagrams of above. Disulfide bonds are annotated as ancestral (found across all genera) or additional (genera/virus specific). **(C)** HCV E1 was mutated to remove each of its 4 disulfide bonds, in turn, or to add additional bonds found in Pegiviruses. Each mutant E1E2 was assessed for expression (either E1 or E2), CD81 receptor binding, E1E2 assembly, and pseudotyped virus infection. Mutated residues are numbered from the start of E1. Data are expressed relative to H77 WT isolate E1E2, mean values $n \geq 3$; error bars indicate standard error. Asterisks denote degree of statistical significance compared to WT (one-way ANOVA). Underlying numerical data are available in S1 File. BVDV, bovine viral diarrhoea virus; HCV, hepatitis C virus; HCV-P, Hepacivirus-P; pFP, putative fusion peptide; SPgV, Simian Pegivirus; TM, transmembrane; WT, wild-type.

beta-sheet sits on top of a distinctive helical hairpin motif (corresponding to the pFP), while the C-terminus is composed of the transmembrane domain and proximal helices. These N and C terminal modules are linked via a loop that bridges around and across the top of the protein. The very N-terminus of E1 forms a structurally divergent tail, which may mediate species-specific interactions with E2 (see below). Notably, in this arrangement, the helical hairpin/pFP is

packed alongside the transmembrane domain, close to, or within, the presumed location of the viral membrane.

E1 is crosslinked by various intramolecular disulfide bonds (Figs 2B and S16), two of which appear to be "ancestral" being present in each genus. One of these ancestral bonds sits across the inward and outward chains that form the first conserved N-terminal beta-strand; the other secures the bridging loop to the central beta-sheet. E1 from Pestiviruses possess only these ancestral intramolecular bonds; some Pestiviruses (e.g., atypical porcine Pestivirus) even lack the second of these bonds. Hepaci- and Pegiviruses have various additional bonds; many of these, however, appear to reiterate the interactions imposed by the ancestral bonds, sitting within and proximal to the first beta-sheet, or linking the bridging loop to the N-terminal region of E1. One additional disulfide bond crosslinks the tip of the helical hairpin/pFP; this appears to be specific to the Hepaciviruses.

The HCV E1 model is consistent with experimental structures of various E1 peptide fragments and, importantly, the recently reported cryoEM model of E1E2 [35,36]; the N-terminal portion of E1, in particular, is in excellent agreement with the cryoEM model, in which the predicted disulfide bonds 1, 2, and 3 are also confirmed (S17A and S17B Fig and S1 Text). However, our model, and the recent cryoEM model, differs significantly from a previous crystal structure of the N-terminal region of E1 (S17C Fig and S1 Text) [35,36]; whether this represents a crystallisation artefact or a functionally relevant alternative conformation requires further investigation. Systematic comparison of our predicted E1 structures with entries in the Protein Data Bank, AlphaFold/EBI, and the ESM Atlas Structure Databases revealed no significant structural similarity, other than with the HCV E1 cryoEM structure [37]. This suggests that E1 exhibits a unique protein fold. Moreover, the lack of similarity between E1 with other viral glycoproteins is consistent with the notion that the Hepaci-, Pegi-, and Pestiviruses may possess a novel fusion mechanism.

We explored the importance of E1 disulfide bonds through mutagenesis and antigenic/functional characterisation in HCV E1E2 pseudotyped viruses (Fig 2C and 2D). Individual loss of any of the 4 HCV E1 disulfide bonds did not negatively impact expression of E1 and E2 or affect binding to the CD81 receptor (which occurs via E2, without contribution from E1). However, loss of C16–C35, an ancestral bond, or C38–C113, an additional bond, prevented E1E2 dimer assembly and pseudotype infection. Notably, C47–C115 (ancestral, connecting the bridging loop and beta-sheet, missing in some Pestiviruses) and C81–C90 (additional, crosslinking the helical hairpin) were not essential for E1E2 dimerisation or infection.

To further explore our models, we mutated HCV E1 to introduce new disulfides, analogous to the additional bonds found in E1 from distantly related Pegiviruses (Fig 2D). R4C-Q111C and E24C I29C (analogous to bonds 1 and 4 in the SPgV E1 model) had limited impacts on E2 expression, CD81 binding, and E1E2 dimerisation; this suggests that the positioning of these bonds is compatible with the predicted topology and structure of E1. R4C Q111C permitted infection, albeit reduced (potentially due to lower levels of E1), whereas E24C I29C was noninfectious despite normal expression/folding; this may indicate that E1E2 are locked in a properly folded, but inactive, state. Note, in vitro experimental methods are provided in S1 Text.

## Structural conservation in E2

The high degree of predicted structural similarity in E1 would suggest that it performs a mechanistically conserved role in membrane fusion, consistent with it possessing the putative fusion peptide [30–34]. In contrast, E2 exhibits high structural divergence between, and even within, viral genera. This, most likely, indicates that E2 mediates species-specific interactions with a particular host; indeed, each genera varies widely in tissue tropism (e.g., Hepaciviruses exclusively

infect the liver, while Pegiviruses target bone marrow). Nonetheless, unbiased structural comparison identified some consistent elements across divergent E2 models (Fig 3A and 3B). Hepaci- and Pegivirus E2 share a beta-sandwich fold with conserved topology and structure (originally reported in the experimentally determined models of HCV [25,26]). This central element appears to act as a scaffold around which various species-specific extensions and loops are arranged. This beta-sandwich is, however, absent from Pestiviruses. Another consistent, yet minimal, feature is an extended beta strand that corresponds to the Back Layer apparent in the crystal structures of HCV E2 [25,38]. This region is important for HCV entry [38] and is present in E2 models across each genus, suggesting mechanistic conservation. Finally, in all cases, the transmembrane domain is predicted to adopt a helical hairpin conformation.

Notably, we also identified structural similarity between E1 and E2 (Fig 3C and 3D). The topology and arrangement of the E1 beta-sheet and helical hairpin (the N-terminal module; described in Fig 2) are mirrored in the C-terminal stem region and transmembrane domain of E2. This is best exemplified by comparison of the E1 and E2 beta-sheet elements across viral species; here, unbiased structural comparison supports similarity between the N-terminal beta-sheet of Pestivirus E1 and the C-terminal beta-sheet of Hepacivirus E2 (Fig 3E—note areas of similarity denoted by orange colouring in the heatmap—and 3F). We propose that this topological/structural similarity may point to the evolutionary origin of E1E2, with an ancient genetic duplication event from a progenitor E1-like fusion protein, giving rise to a proto-E2 molecular partner, analogous to the domain duplication event evident in the N- and C-terminal domains of retroviral capsids [39,40]. The E1E2 of Hepaci-/Pegi-/Pestiviruses would have all arisen from this common ancestor. In this scheme, E1 may be responsible for the primary fusion activity and, therefore, has a conserved structure, whereas E2 performs a regulatory role and has been free to evolve structural elaborations that permit diversification of host and tissue range.

## The E1E2 interface

It has long been established that the folding and activity of E1E2 are interdependent, and we expect conserved intermolecular communications to regulate and mediate membrane fusion. Our models indicate that E1 wraps tightly around the transmembrane domain and stem regions at the C-terminus of E2 with an interface composed of multiple E1 regions (Figs 4A, 4B, and S18 and S1 Text). For example, the E2 transmembrane domain packs against the helical hairpin/pFP of E1, while the E2 stem region abuts the bridging loop of E1. These various interactions create an "ancestral" interface found in all E1E2 (Fig 4C) and are consistent with the "stem in hand" model proposed by Torrents de la Peña and colleagues [35] when describing their recent structure of HCV E1E2. Notably, this ancestral interaction is centred around the region of E2 that, we propose, originated through genetic duplication of E1 (Fig 3C).

Beyond the ancestral interface, there are additional genus/species-specific interactions. Unique to Pestiviruses is a predicted E1E2 intermolecular disulfide bond formed between a beta-hairpin, found directly upstream of the transmembrane proximal region in E1, and a loop extending from the beta-sheet element in the stem of E2 (Fig 4C and 4D). Notably, this region of E2 forms the interface of a disulfide stabilised E2–E2 homodimer in BVDV E2 crystal structures [23,24]; whether a switch from E1E2 heterodimers to E2E2 homodimers [41] is necessary for fusion requires further investigation. In Hepaci- and Pegiviruses, the very N-terminus of E1 varies in length and conformation between species. This N-terminal tail makes further contacts with species-specific loops extending from the base of E2; this mode of interaction is largely missing from Pestiviruses, where the N-terminal extension is absent from E1 (Fig 4E).

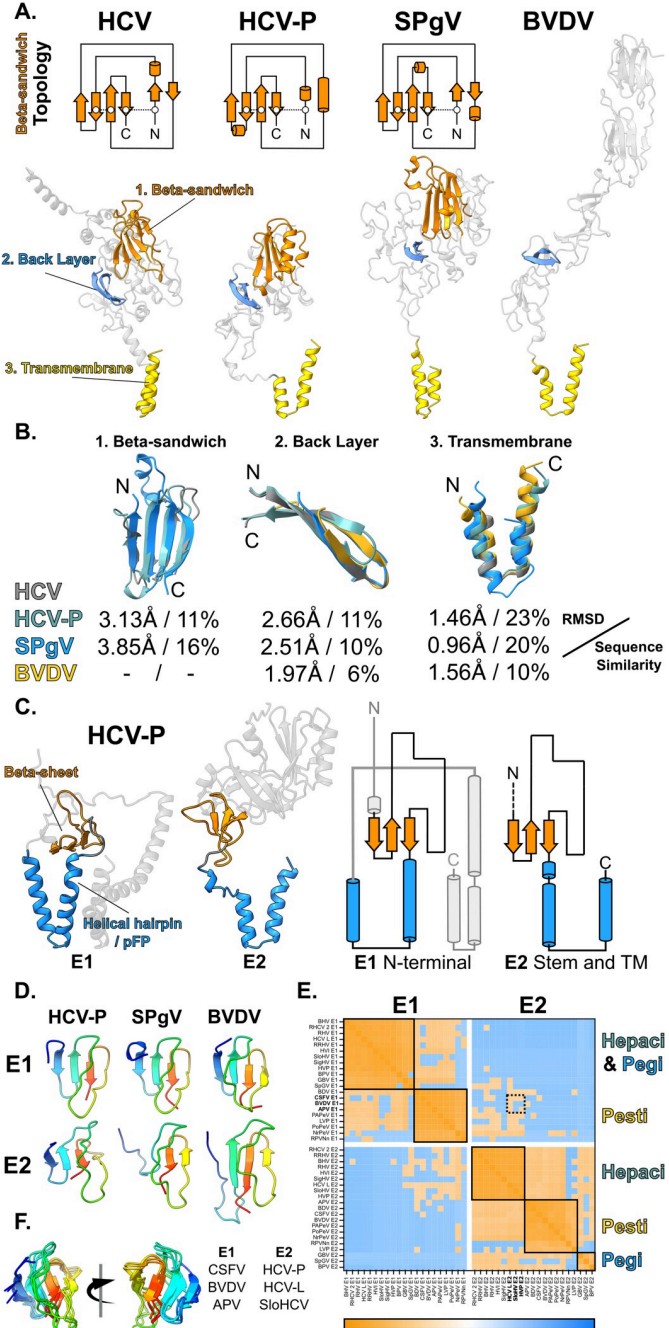

**Fig 3. Structural conservation in E2. (A)** Bottom, ribbon diagrams of predicted E2 structures for HCV, HCV-P, SPgV, and BVDV; consistent features are colour coded as shown on the HCV structure, left. Top, topology diagrams for the conserved beta-sandwich scaffold found in all Hepaci- and Pegiviruses. **(B)** Ribbon diagrams of overlaid features for HCV, HCV-P, SPgV, and BVDV are accompanied by RMSD and sequence similarity values by comparison to HCV. **(C)** Structural conservation between the N-terminus of E1 and the C-terminal stem and transmembrane domain of E2. Ribbon (left) and topology (right) diagrams of predicted HCV-P E1 and E2 structures highlighting similarity. **(D)** Ribbon diagrams (rainbow colour coded blue to red, N to C terminus) of beta-sheet feature conserved across E1 and E2 in HCV-P, SPgV, and BVDV. **(E)** Structural similarity heatmap for Hepaci-, Pegi-, and Pestivirus E1 and E2 beta-sheet elements (as shown in D); orange denotes low distance and, therefore, high similarity. **(F)** Structural conservation between Pestivirus E1 and Hepacivirus E2 beta-sheets demonstrated by superposition of stated structures (annotated as dashed box on E; a full list of viruses, their accession numbers, and abbreviations are provided in S2 File). Structures are shown in 2 opposite orientations, rainbow colour coding from N to C termini. Underlying numerical data are available in S1 File. APV, Aydin-like Pestivirus; BVDV, bovine viral diarrhoea virus;

CSFV, classical swine fever virus; HCV, hepatitis C virus; HCV-L, Hepacivirus L; HCV-P, Hepacivirus-P; RMSD, root mean square deviation; SloHCV, Sloth Hepacivirus; SPgV, Simian Pegivirus.

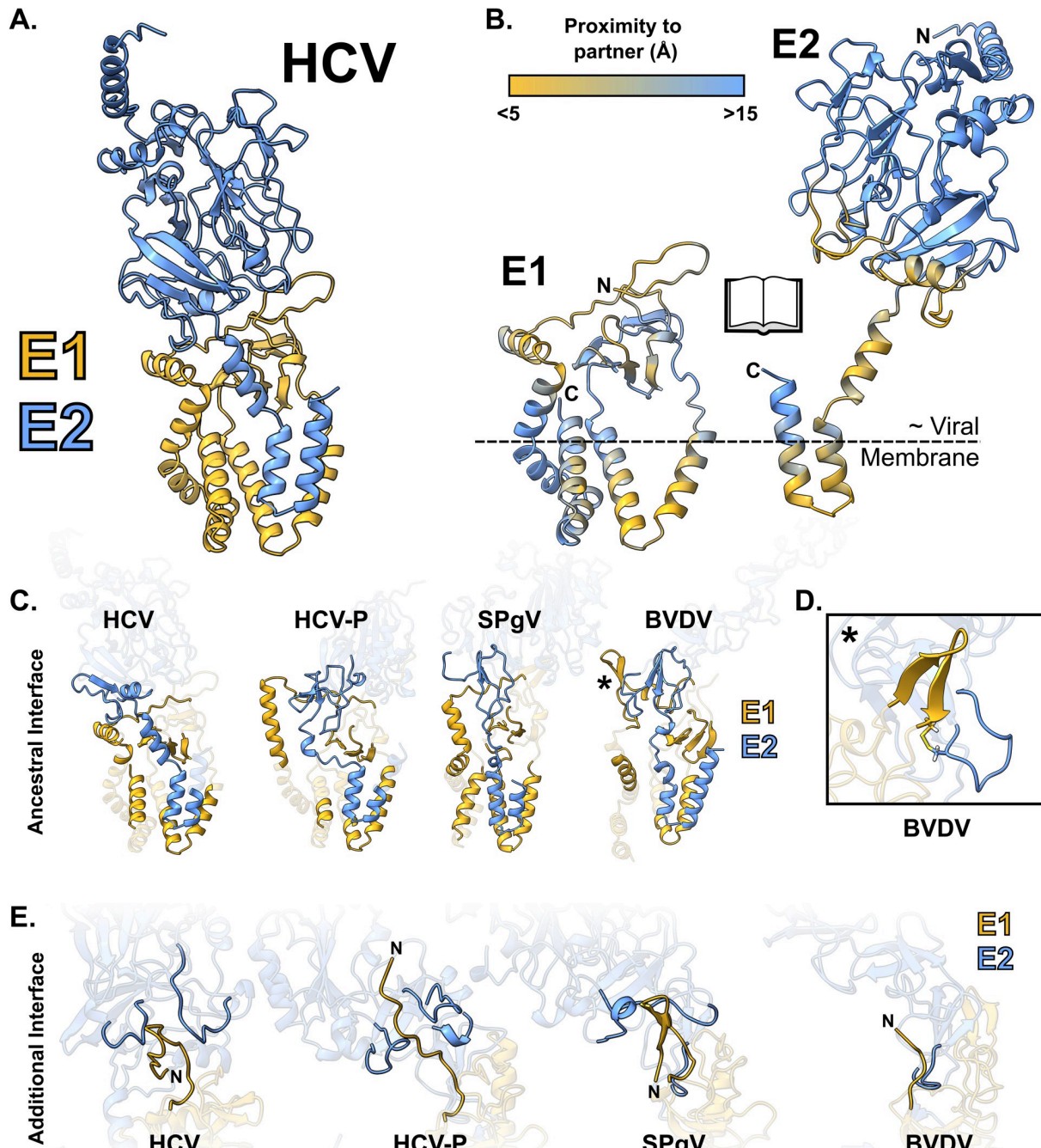

**Fig 4. The E1E2 interface.** (**A**) Ribbon diagram of HCV E1E2 complex. (**B**) HCV E1E2 complex interaction interface in an "open book" representation. Residues are colour coded by their shortest distance to the partner protein (Cα to Cα) as indicated by the colour key. The approximate location of the outer leaflet of the viral membrane is inferred by the positions of the E1 and E2 transmembrane domains. (**C**) Ribbon diagrams illustrating the "ancestral" E1E2 interface found in all genera; regions of contact/high proximity have been highlighted. (**D**) Pestiviruses possess an E1E2 intermolecular disulfide bond, the position of which is marked by an asterisk on (**C**). (**E**) Ribbon diagrams illustrating the "additional" species-specific interface between the extended tail of E1, found in Hepaci- and Pegiviruses, and loops at the base of E2. BVDV, bovine viral diarrhoea virus; HCV, hepatitis C virus; HCV-P, Hepacivirus-P; SPgV, Simian Pegivirus.

## Discussion

We have combined systematic curation of viral genome sequences with AlphaFold modelling to perform a structural survey of the E1E2 glycoproteins from 60 diverse viruses and provide models for 24 E1E2 complexes. These analyses revealed that the E1 protein from Hepaci-, Pegi-, and Pestiviruses exhibits a unique but very consistent structure; this suggests that these viruses share a potentially novel viral membrane fusion mechanism.

We have high confidence that these predicted structures are informative: AlphaFold performed very well in benchmarking tests (S1 Fig and S1 Text), our predictions yield very consistent E1 structures from completely independent datasets (Fig 1), and our models compare favourably to the current best understanding of E1 and E1E2 based on experimental structures (S17 Fig and S1 Text). There will, undoubtedly, be some inaccuracies in our models. However, insights gained from the depth and breadth of structural comparisons in this study would be difficult to achieve with classical structural biology.

A recurrent theme in our analysis is that of "ancestral" features of E1E2, found in each genus, augmented with "additional" genus or species-specific elaborations. Delineating the contributions made by "ancestral" elements is likely to provide fundamental mechanistic insights. In this respect, Pestivirus E1, with only 1 or 2 intramolecular disulfide bonds and lacking an N-terminal extension, may represent the simplest iteration of this fusion mechanism. This would also be consistent with the ancient ancestral status of Pestiviruses apparent in phylogenetic analyses [42].

Ultimately, while this work provides the first evidence of a potentially novel fusion mechanism, there are many important questions left to answer. For example, we do not know whether AlphaFold delivers a pre- or postfusion structure (or some intermediate form). However, the agreement of our models with various experimental structures of E2 and E1E2, which required protein stabilisation with neutralising antibodies [25,26,35,43], supports the notion of a prefusion state.

A full understanding of how E1E2 performs membrane fusion will require a detailed mapping of structure to function and an appreciation of the necessary conformational rearrangements. Nonetheless, we can consider our findings in the context of various proposed models of Hepaci- and Pestivirus fusion. By analogy to other well-described fusion mechanisms, we would expect either E1 or E2 to be the primary fusogen, with a hydrophobic peptide capable of inserting into host membranes; there remains significant debate in this area [24,44–46]. E2 from both HCV and BVDV possess hydrophobic regions that are candidate fusion peptides. However, we observed significant heterogeneity in E2 length, sequence, and structure, such that it is difficult to identify corresponding hydrophobic regions across all species; we would expect strict conservation of a mechanistically essential fusion peptide. This would suggest that E2 is not the primary fusogen.

On the contrary, E1 is structurally conserved and a putative fusion peptide has been identified in HCV (corresponding to the helical hairpin found in all species). This would argue in favour of E1 being the primary fusogen. However, based on our structures and recent cryoEM data [35], the helical hairpin/pFP is packed closely with the E1 and E2 transmembrane domains and may be buried within the viral membrane. Consistent with this, the pFP can act as a membrane anchor for E1 if the transmembrane domain is deleted [47]. Extraction of the pFP from this environment, such that it can be inserted into a host membrane, would be thermodynamically unfavourable. E1 is also small compared to class I-III fusogens (<200 residues, approximately 6.5 nm total height in our structures) and would have to undergo extensive refolding to bridge the gap between viral and host membrane (approximately 20 nm). We should, therefore, consider that E1E2 does not function in an analogous manner to known

fusion machinery. Other uncharacterised viral fusion machinery, such as the surface glycoprotein of hepatitis B virus, also possess multiple transmembrane domains and no obvious fusion peptide [48]. In these cases, we cannot exclude the possibility that membrane fusion is achieved without insertion of a viral fusion peptide into the target host membrane or that it requires the concerted action of multiple proteins.

Aside from the precise mechanism of fusion, we can also consider the means by which E1E2 sense the molecular cues that trigger their activity. Hepaci- and Pestiviruses are known to require acidic activation to achieve entry from the endosome, but prior events, such as receptor engagement, are necessary to prime pH sensitivity [49–51]. Due to its intermediate pKa value, histidine functions as a pH sensor in many viral fusion proteins [52]. We may expect histidine residues fulfilling this role in E1E2 to be highly conserved, and yet no single histidine residue is consistently found across all species. Thus far, receptor engagement (i.e., CD46 in BVDV and CD81/SR-B1 in HCV [53–55]) is only known to occur via E2; once again, the sequence and structural divergence of E2 makes it difficult to identify commonalities that may link receptor engagement with E1E2 fusion activity. Although, the cross-genera structural conservation in the C-terminus of E2 (back layer and onward; Fig 3) may suggest that species-specific receptor interactions are communicated to this region, which is in close proximity to E1, providing the possibility for intermolecular crosstalk. Notably, the only residues conserved across all species are the E1 cysteine residues that form the "ancestral" disulfide bonds (Fig 2). There is evidence that disulfide isomerisation may be important for Hepaci- and Pestivirus entry [23,24,51,56,57]. Therefore, some of these ancestral bonds may be broken and/or shuffled during the conformational rearrangements necessary for fusion.

Finally, there remain overarching questions around the evolution of membrane fusion in these viruses. Structural/topological similarity between E1 and E2 (Fig 3) may suggest that E1E2 originated in an ancient genetic duplication event; might a primordial fusion apparatus only containing E1 exist elsewhere in the virome or beyond? Also, how does Hepaci-/Pegi-/Pestivirus evolutionary history relate to the *Orthoflavivirus* genus (distant ancestors, also in the Flaviviridae family), which exhibits a class-II membrane fusion protein? Did these genera diverge from one another due to the acquisition/emergence of alternative membrane fusion mechanisms?

We expect the structural insights gained through this study will provide a platform for wide-ranging investigations on these questions, and further our understanding of viral membrane fusion mechanisms. This will not only deliver fundamental knowledge but also guide structure-based design of antiviral interventions and vaccines.

## Supporting information

**S1 Text. A detailed account of the AlphaFold strategy used in this work and the associated quality control assessments.** Detailed in vitro experimental methods are also provided. (DOCX)

**S1 File. Excel spreadsheet containing the underlying numerical data for Figs 1B, 1E, 2D, 3E, S1B, S1C, S1D, S3, S10, S14, S15, and S18C.** (XLSX)

**S2 File. Excel spreadsheet summarising the viral species used in this study, their sequence accession numbers, and abbreviated names.** (XLSX)

**S1 Fig. Benchmarking of AlphaFold against relevant viral targets. (A)** AlphaFold predicted structures superposed with their cognate experimental structure from the protein database

(PDB codes are provided for each structure; TBEV, tick-borne encephalitis virus). In each case, the lower structure represents the AlphaFold model colour coded by RMSD from its experimentally determined partner structure. Blue indicates disagreement, as denoted in the colour key. (**B**) MolProbity scores of all AlphaFold benchmarking models before and after AMBER relaxation. Each data point represents an individual model, $n = 50$ (5 candidate models per viral target). Lower values indicate higher model quality. (**C**) AlphaFold model confidence plotted against MolProbity score for all benchmarking structures ($n = 50$, with rank 1 models shown in blue). The negative correlation indicates higher model quality with higher confidence scores (linear correlation). (**D**) Comparison of MolProbity scores for AlphaFold structures and their cognate experimental structures. Asterisks indicate degree of statistical significance (*t* test). Underlying numerical data are available in S1 File. Further description can be found in S1 Text.
(TIF)

**S2 Fig. Hepacivirus NS5B phylogenetic tree.** Subclades are numbered and colour coded; species in bold text represent clade-specific reference viruses. E1 and E2 annotations from these reference viruses were propagated throughout aligned whole genome sequences of each subclade. Further description can be found in S1 Text.
(TIF)

**S3 Fig. Protein lengths of E1 and E2 from diverse Hepaci-, Pegi-, and Pestiviruses.** Each data point represents an individual viral species, $n = 32$, 15, and 13, respectively. Underlying numerical data are available in S1 File. Further description can be found in S1 Text.
(TIF)

**S4 Fig. Hepacivirus E1 monomer AlphaFold models.** Residues are colour coded by pLDDT prediction confidence. Models are arranged in order of descending prediction confidence. Further description can be found in S1 Text.
(TIF)

**S5 Fig. Pegivirus E1 monomer AlphaFold models.** Residues are colour coded by pLDDT prediction confidence. Models are arranged in order of descending prediction confidence. Further description can be found in S1 Text.
(TIF)

**S6 Fig. Pestivirus E1 monomer AlphaFold models.** Residues are colour coded by pLDDT prediction confidence. Models are arranged in order of descending prediction confidence. Further description can be found in S1 Text.
(TIF)

**S7 Fig. Hepacivirus E2 monomer AlphaFold models.** Residues are colour coded by pLDDT prediction confidence. Models are arranged in order of descending prediction confidence. Further description can be found in S1 Text.
(TIF)

**S8 Fig. Pegivirus E2 monomer AlphaFold models.** Residues are colour coded by pLDDT prediction confidence. Models are arranged in order of descending prediction confidence. Further description can be found in S1 Text.
(TIF)

**S9 Fig. Pestivirus E2 monomer AlphaFold models.** Residues are colour coded by pLDDT prediction confidence. Models are arranged in order of descending prediction confidence.

Further description can be found in S1 Text.
(TIF)

**S10 Fig. Model quality metrics for E1 and E2 monomer models.** pLDDT prediction confidence (upper plots) and MolProbity scores (lower plots) for E1 and E2 models from diverse Hepaci-, Pegi-, and Pestiviruses. Each data point represents an individual viral species, $n$ = 32, 15, and 13, respectively. Dashed line on upper right plot indicates pLDDT = 70 cutoff that was used to select viruses for modelling of E1E2 complexes. Underlying numerical data are available in S1 File. Further description can be found in S1 Text.
(TIF)

**S11 Fig. Hepacivirus E1E2 complex AlphaFold models.** Residues are colour coded by pLDDT prediction confidence. Models are arranged in order of descending prediction confidence. Further description can be found in S1 Text.
(TIF)

**S12 Fig. Pegivirus E1E2 complex AlphaFold models.** Residues are colour coded by pLDDT prediction confidence. Models are arranged in order of descending prediction confidence. Further description can be found in S1 Text.
(TIF)

**S13 Fig. Pestivirus E1E2 complex AlphaFold models.** Residues are colour coded by pLDDT prediction confidence. Models are arranged in order of descending prediction confidence. Further description can be found in S1 Text.
(TIF)

**S14 Fig. Model quality metrics for E1 and E2 complex models.** pLDDT prediction confidence (upper plots) and MolProbity scores (lower plots) for E1 and E2 modelled as a monomer or in a complex. Each data point represents an individual viral species, $n$ = 24. Asterisks indicate degree of statistical significance ($t$ test). Underlying numerical data are available in S1 File. Further description can be found in S1 Text.
(TIF)

**S15 Fig. Unbiased structural comparison and clustering of E1 and E2 models.** All against all comparison was performed on either E1 or E2 (derived from E1E2 complex models) using the DALI server. The heat map (right) provides pairwise distances indicating structural similarity (orange denoting low distances and, therefore, high similarity). Structures are also clustered by similarity, as denoted by the dendogram (left). Boxes are drawn to group structures with high similarity. The position of HCV is additionally highlighted. Underlying numerical data are available in S1 File. Further description can be found in S1 Text.
(TIF)

**S16 Fig. E1 intramolecular disulfide bonds. (A)** Ribbon diagrams illustrating the location of the ancestral and additional disulfide bonds described in Fig 2. **(B)** Linear representation of HCV E1 sequence (H77 isolate), annotated with the position of each disulfide bond.
(TIF)

**S17 Fig. Comparison of HCV AlphaFold E1 model with experimentally determined structures. (A)** E1 model compared to cognate cryoEM (PDB 7T6X) and peptide structures (2KNU and 4N0Y); superposition was achieved by alignment of structure and sequence. **(B)** Comparison of predicted E1E2 disulfide bonds with 3 experimentally determined structures. Bonds in black text are in agreement; red text indicates disagreement. Dashes indicate disulfides that were absent from the protein construct; NR, not resolved. **(C)** The N-terminal portion of

AlphaFold HCV E1 model and the cognate crystal structure (PDB:4UOI). Further description can be found in S1 Text.
(TIF)

**S18 Fig. Comparison of the E1E2 interface in Hepaci-, Pegi-, and Pestiviruses. (A)** Ribbon diagrams of the E1E2 complex from HCV, HCV-P, SPgV, and BVDV. (**B**) HCV E1E2 complex interaction interface in an "open book" representation. Residues are colour coded by their shortest distance to the partner protein (Cα to Cα) as indicated by the colour key. Symbols annotate contact sites between E1 and E2. The approximate location of the outer leaflet of the viral membrane is inferred by the positions of the E1 and E2 transmembrane domains. (**C**) Plots provide E1 and E2 per residue shortest distance to partner protein (i.e., proximity of any given E1 residue to E2, and vice versa) for HCV, HCV-P, SPgV, and BVDV. Symbols relate to sites of contact, as annotated in (**B**). Lines are colour coded by structural features as in main text 2A (E1) and 3C (E2). Underlying numerical data are available in S1 File. Further description can be found in S1 Text.
(TIF)

## Acknowledgments

This work draws upon many thousands of publicly available viral sequences derived from clinical samples. We, therefore, thank the Unseen Hands project and Terrence Higgins Trust Scotland as representatives of people living with blood-borne viruses who have generously donated blood samples for biomedical research.

## Author Contributions

**Conceptualization:** Joe Grove.

**Data curation:** Michael R. Oliver, Robert J. Gifford, Joe Grove.

**Formal analysis:** Michael R. Oliver, Joe Grove.

**Funding acquisition:** Joe Grove.

**Investigation:** Michael R. Oliver, Kamilla Toon, Charlotte B. Lewis, Stephen Devlin, Robert J. Gifford, Joe Grove.

**Methodology:** Kamilla Toon, Robert J. Gifford, Joe Grove.

**Project administration:** Joe Grove.

**Supervision:** Joe Grove.

**Visualization:** Joe Grove.

**Writing – original draft:** Joe Grove.

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
