## [Editor Report · Decision Letter 0]

2 Feb 2023

Dear Joe, 

Thank you for submitting your manuscript entitled "Evidence of a novel viral membrane fusion mechanism shared by the Hepaci, Pegi and Pestiviruses." for consideration as a Research Article by PLOS Biology.

Your manuscript has now been evaluated by the PLOS Biology editorial staff, as well as by an academic editor with relevant expertise, and I am writing to let you know that we would like to send your submission out for external peer review.

Once your full submission is complete, your paper will undergo a series of checks in preparation for peer review. After your manuscript has passed the checks it will be sent out for review. To provide the metadata for your submission, please Login to Editorial Manager (https://www.editorialmanager.com/pbiology) within two working days, i.e. by Feb 04 2023 11:59PM.

Kind regards,

Paula

---

Senior Editor

PLOS Biology

---

## [Decision Letter · Decision Letter 1]

15 Mar 2023

Dear Dr. Grove,

Thank you for your patience while your manuscript "Evidence of a novel viral membrane fusion mechanism shared by the Hepaci, Pegi and Pestiviruses." went through peer-review at PLOS Biology. Your manuscript has now been evaluated by the PLOS Biology editors, an Academic Editor with relevant expertise, and by several independent reviewers.

In light of the reviews, which you will find at the end of this email, we are pleased to offer you the opportunity to address the comments from the reviewers in a revision that we anticipate should not take you very long. We will then assess your revised manuscript and your response to the reviewers' comments with our Academic Editor aiming to avoid further rounds of peer-review, although might need to consult with the reviewers, depending on the nature of the revisions.

In particular, it is important that you tone down any claims of fusion mechanism, the manuscript would need to add more discussion along potential fusion models and it would need to be refocused to discuss the issues raised by reviewer #2.

Please also address the following formatting and editorial issues:

1. DATA POLICY:

A) Supplementary files (e.g., excel). Please ensure that all data files are uploaded as 'Supporting Information' and are invariably referred to (in the manuscript, figure legends, and the Description field when uploading your files) using the following format verbatim: S1 Data, S2 Data, etc. Multiple panels of a single or even several figures can be included as multiple sheets in one excel file that is saved using exactly the following convention: S1_Data.xlsx (using an underscore).

B) Deposition in a publicly available repository. Please also provide the accession code or a reviewer link so that we may view your data before publication. 

Regardless of the method selected, please ensure that you provide the individual numerical values that underlie the summary data displayed in the following figure panels as they are essential for readers to assess your analysis and to reproduce it: Figure 1BDE, 2D, 3E, and Supplementary Figure S1BCD, S3, S10, S14, S15, S17C.

2. We suggest a change in the title to tone down the claims reagrding the mechanism. We suggest: "Common origin and structural features of the envelope proteins of Hepaci, Pegi and Pestiviruses suggests a new membrane fusion mechanism" or "Common origin and structural features of the envelope proteins of Hepaci, Pegi and Pestiviruses suggest a potentially new membrane fusion mechanism".

**IMPORTANT - SUBMITTING YOUR REVISION**

*Resubmission Checklist*

*Published Peer Review*

*PLOS Data Policy*

*Blot and Gel Data Policy*

Sincerely,

Paula

---

Senior Editor

PLOS Biology

REVIEWS:

Reviewer #1: Flavivirus and viral fusion proteins.

Reviewer #2: Structural virology.

Reviewer #1: In this study, the authors look to shed light on the fusion mechanisms of Hepaci, Pegi, and Pestiviruses, an important question in the field. They use Alphafold to model the E1 and E2 glycoprotein structures of 60 different Hepaci, Pegi, and Pestiviruses and draw several conclusions. They find conserved structures in the E1 glycoprotein between viruses as well as ancestral and additional disulfide bonds. They show that several of the predicated disulfide bounds are important for HCV E1E2 assembly and infection. They conclude that the E1 conservation is due to the conserved function of membrane fusion. Last, they find that E2 is not well conserved between viruses, which they predict may be due to differences in virus life cycle and receptor binding. Overall, this is an interesting study that highlights the potential for a new mechanism of membrane fusion for these viruses, distinct from other fusion proteins. However, there are a few concerns that should be addressed for publication.

1. My major concern is that the authors state a number of times (even in the title) that this identifies a "novel membrane fusion mechanism". While this may turn out to be true, this study does not show anything regarding fusion mechanisms. I would say this study identifies the "potential" for a novel membrane fusion mechanism and even new class of fusion glycoprotein. Or maybe "evidence of a unique viral membrane fusion mechanism" . It is clear from your modeling these don't look like the other fusion proteins. However, like the authors said there are many caveats here (pre- vs. post-fusion structures, limited experiments). Therefore, I suggest the authors reword these phrases to reflect their conclusions. Regardless, highlighting that these may be a new class of fusion protein is really cool!

If the authors want to go down the road of fusion, can the authors speculate on a model of how these proteins would initiate membrane fusion? There is significant work done on HCV pH-dependent entry. Can you use your modeling to fit into these studies? In addition, there are proposed pestivirus models of fusion (PMID: 23569276). How do your models fit in there? Are there flexible domains that could move to drive fusion? I think speaking more about fusion mechanisms will be required to keep the "novel membrane fusion mechanism" angle. 

2. A more clarity comment and I'm sorry if I missed it. Can you provide on Figure 1 and 2 where the fusion peptide is located? Also, in figure 2A. How many transmembrane domains are present? By this schematic, it looks like the blue helices are TMD too. Is that true? Can you modify to clear that up?

Reviewer #2: The manuscript by Oliver et al. reports a very interesting study, revealing important properties of the envelope glycoproteins proteins E1 and E2 of viruses in three of the four genera of the Flaviviridae family. They present a phylogenetic analysis together with AlphaFold structural predictions showing that E1 and E2 are related and appear to have evolved by an ancient gene duplication event. Furthermore, they reveal that E2 has diverged much more than E1 across the different genera and even within a genus. Previous structural studies had shown that E1 and E2 are unrelated to their counterparts in members of the Flavivirus genus (the fourth genus in the family), which have acquired a fusion protein with homologs in viruses from unrelated families, such as alphaviruses and bunyaviruses. The value of the present study is that it now shows that the three remaining genera of the Flaviviridae (hepaciviruses, pegiviruses and pestiviruses) share a common ancestor for their envelope proteins. In addition, the paper highlights additional information about this class of envelope proteins that the authors do not address in their current manuscript.

This last point concerns the trans-membrane (TM) regions. Because most of the viral fusion proteins are type 1 transmembrane proteins anchored at the viral surface by a single TM segment at the C-terminal end, it had been expected that E1 and E2 would be the same. The Flaviviruses are actually an exception, because the envelope proteins in this genus have an alpha-helical segment instead of a single TM segment at the C-terminal end, unlike their counterpart in alphaviruses and bunyaviruses. The AlphaFold predictions indicate instead that E1 and E2 span the membrane multiple times. This is very clear in Figure 4A, showing the TM helical bundle anchored in the membrane, and the beta-sheet and additional extensions projecting out of the bundle to the extra-viral space. The authors, however, appear to overlook this important result. Furthermore, as they show in Figure 4D, the ancestor of these proteins appears to have been a simple protein with a small beta sheet followed by a hydrophobic alpha-helical hairpin, which can be safely interpreted as a TM anchor. E1 and E2 thus appear to have undergone different paths during their subsequent evolution, from a potential homodimer to then a heterodimer (upon gene duplication). E1 acquired an additional alpha-helical TM hairpin and peri-membrane-region at its C-terminal half, and E2 evolved host-specific extensions and additions to the initial beta-sheet, to the point that the beta sheet element is altogether absent in the pestiviruses. Furthermore, the distance in the membrane spanned by the TM segments is shorter than a normal lipid bilayer spanning segment, but this is also the case for the alpha-helical TM hairpin of their fusion protein. The reason is that the bilayer in the viral particle is highly constrained and very thin in the areas where the TM segments are inserted, as shown by the cryo-EM structures. For the hepatitis C virus, in contrast, it has been proposed that the glycoproteins are anchored in a lipid structure similar to that of lipid droplets, which display a glycerophospholipid monolayer surrounding t triglyceride-rich core. The dimension of the structure displayed in Figure 4D would be an excellent fit on the monolayer in this model.

The original postulate that the first hydrophobic segment of E1 could constitute a fusion loop was done at a time where no structural data were available, and when it was considered that hepaciviruses, pegiviruses and pestivirus would function similarly to the fusion proteins of viruses that had already been characterized, and which all feature a fusion loop (or fusion peptide) that inserts into the target membrane during a conformational change. But there are other viruses, such as the hepadnaviruses (HBV, for instance), the envelope protein of which has four TM segments and no predicted fusion loop or peptide. Or the poxviruses, which have a membrane fusion complex composed of at least 10 different proteins, all with transmembrane regions. It is evident that these viruses cannot induce membrane fusion using the mechanism of those for which the structural studies are available. This study therefore opens the door for beginning to understand the mechanism of fusion developed by them. It could be a concerted action between the fusion proteins, not necessarily the action of one of the two partners of the heterodimer, like in the class II fusion proteins.

In my view, therefore, the authors are right to discuss that E1 and E2 must display a different mechanism to induce fusion than the flaviviruses, but they fail to see that they have provided crucial evidence to lump these three genera into a broader context of viruses having multiple TM segments and for which the mechanism of action is not understood. And here is where I see the true value of this paper.

It is therefore important that the authors revisit the interpretation of their results, which are indeed major. The study was very carefully done, with an extensive phylogenetic analysis and a very detailed description of the way AlphaFold was used to obtain the results. In particular, the Venn diagrams showing no overlap between the sequences used in the multiple-sequence alignments used by ALphaFold to reach the predictions for different genera. Unfortunately, all this part is in the supplementary materials, and would belong in the main text, even if most of the supplementary Figures can remain so. The authors also have made a functional analysis knocking out the disulfide bonds, which adds to the significance of the study. 

In addition to my major comments above, I have additional minor comments concerning the manuscript as presented, which should also help improve it. These comments are listed essentially in the order as the issues appear in the text, and not necessarily in a logical way.

1. Please provide more background concerning HCV-C, HCV-P, etc. The average reader of PLoS biology is most likely not a specialist in hepatitis C virus, and this terminology is certainly unfamiliar.

2. Maybe draw the disulfide bonds as green sticks in the cartoons of Fig. 2B

3. Line 30: please add bunyaviruses to the list of viruses with class II fusion proteins.

4. Line 33: "eliminated any genetic homology". The term "homology" is incorrectly used here. What the authors mean is that the homology cannot be detected by sequence comparisons, as during divergent evolution the proteins have lost all sequence similarity. But their homology can still be detected by structural comparisons. So, what the authors mean is that evolutionary divergence has eliminated any sequence similarity that could allow to detect homology. This is also true for the supplementary text, where they say: "this revealed low homology across most of the genome in every genus". This sentence does not make sense, as obviously they are clear homologs (in line 73 of the supplementary material). The same is true for the sentence in line 144, where the authors mean "the high degree of structural similarity". Homology cannot have a high or low degree: it is binary. Two proteins are either homologous or they are not, what differs is the degree of diversity from their common ancestor.

5. Line 50: the minor glycoprotein E1. It is not a "minor" protein, as it is produced in stoichiometric amounts with E2. 

6. Line 69: primary sequence similarity, not homology. Same for line 72: little/no sequence similarity, not homology.

7. Line 186: should be "HCV-C E1E2"

8. Legend to Fig. 3E: "Structures are shown in two opposite orientations". This is not clear: It seems to me that they are viewed rotated by 180 degrees about a vertical axis. It would be helpful to add a labelled rotation symbol in between the two views.

9. Figure 4B: Is this an "open book" representation of the complex shown in 4A? Please state so in the legend (maybe also provide an open book symbol, for clarity). Same for Fig. S17B.

10. Legend to Figure 4C: "Ribbon diagrams demonstrating the 'ancestral' E1E2 interface". I would say "illustrating" rather than "demonstrating the 'ancestral' E1E2 interface". Same for 4E.

11. SM lines 114-115: "A simple visual examination of the Hepaci and Pegivirus E1 structures (Fig. S3 and 4)" Should be Fig. S4 and 5 instead?

12. SM lines 130-131 "such that Pestivirus E1, form E1E2 complexes" there is a typo and an extra comma here, which makes the sentence difficult to read. 

13. SM line 214: typo in "Mutant E12 constructs"? 

14. It might make sense to try the HorA (http://prodata.swmed.edu/horaserver; doi: 10.1093/nar/gkp328) server to assess the homology of E1 and E2, and across the three genera being examined.

---

## [Editor Report · Decision Letter 2]

26 May 2023

Dear Joe,

Thank you for the submission of your revised Discovery Report "Structures of the Hepaci-, Pegi- and Pestiviruses envelope proteins suggest a novel membrane fusion mechanism." for publication in PLOS Biology. On behalf of my colleagues and the Academic Editor, Eugene Koonin, I am pleased to say that we can in principle accept your manuscript for publication, provided you address any remaining formatting and reporting issues. These will be detailed in an email you should receive within 2-3 business days from our colleagues in the journal operations team; no action is required from you until then. Please note that we will not be able to formally accept your manuscript and schedule it for publication until you have completed any requested changes.

PRESS

Sincerely, 

Paula Jauregui

---

Senior Editor

PLOS Biology
